# Triage of Patients Suspected of COVID-19 in Chronic Hemodialysis: Eosinophil Count Differentiates Low and High Suspicion of COVID-19

**DOI:** 10.3390/jcm10010004

**Published:** 2020-12-22

**Authors:** Romain Vial, Marion Gully, Mickael Bobot, Violaine Scarfoglière, Philippe Brunet, Dammar Bouchouareb, Ariane Duval, He-oh Zino, Julien Faraut, Océane Jehel, Yaël Berdad-Haddad, Stéphane Burtey, Pierre-André Jarrot, Guillaume Lano, Thomas Robert

**Affiliations:** 1Centre of Nephrology and Renal Transplantation, Hôpital de la Conception, CHU de Marseille, 13005 Marseille, France; Romain.VIAL@ap-hm.fr (R.V.); marion.gully@ap-hm.fr (M.G.); mickael.bobot@ap-hm.fr (M.B.); violaine.scarfogliere@ap-hm.fr (V.S.); philippe.brunet@ap-hm.fr (P.B.); dammar.bouchouareb@ap-hm.fr (D.B.); Heoh.zino@ap-hm.fr (H.-o.Z.); julien.faraut@ap-hm.fr (J.F.); oceane.jehel@ap-hm.fr (O.J.); stephane.burtey@ap-hm.fr (S.B.); guillaume.lano@ap-hm.fr (G.L.); 2C2VN, Aix-Marseille University, INSERM 1263, INRAe, 13005 Marseille, France; Pierre.JARROT@ap-hm.fr; 3Association des Dialysés Provence et Corse, 13009 Marseille, France; Ariane.DUVAL@ap-hm.fr; 4Hematology Laboratory, Hôpital de la Conception, CHU de Marseille, 13005 Marseille, France; Yael.BERDA@ap-hm.fr; 5Department of Internal Medicine and Clinical Immunology, CHU de Marseille, Hôpital de la Conception, 13005 Marseille, France; 6MMG, Bioinformatics & Genetics, Aix-Marseille Université, UMR_S910, 13004 Marseille, France

**Keywords:** hemodialysis, COVID-19, eosinophil

## Abstract

Background: Daily management to shield chronic dialysis patients from SARS-CoV-2 contamination makes patient care cumbersome. There are no screening methods to date and a molecular biology platform is essential to perform RT-PCR for SARS-CoV-2; however, accessibility remains poor. Our goal was to assess whether the tools routinely used to monitor our hemodialysis patients could represent reliable and quickly accessible diagnostic indicators to improve the management of our hemodialysis patients in this pandemic environment. Methods: In this prospective observational diagnostic study, we recruited patients from La Conception hospital. Patients were eligible for inclusion if suspected of SARS-CoV-2 infection when arriving at our center for a dialysis session between March 12th and April 24th 2020. They were included if both RT-PCR result for SARS-CoV-2 and cell blood count on the day that infection was suspected were available. We calculated the area under the curve (AUC) of the receiver operating characteristic curve. Results: 37 patients were included in the final analysis, of which 16 (43.2%) were COVID-19 positive. For the day of suspected COVID-19, total leukocytes were significantly lower in the COVID-19 positive group (4.1 vs. 7.4 G/L, *p* = 0.0072) and were characterized by lower neutrophils (2.7 vs. 5.1 G/L, *p* = 0.021) and eosinophils (0.01 vs. 0.15 G/L, *p* = 0.0003). Eosinophil count below 0.045 G/L identified SARS-CoV-2 infection with AUC of 0.9 [95% CI 0.81—1] (*p* < 0.0001), sensitivity of 82%, specificity of 86%, a positive predictive value of 82%, a negative predictive value of 86% and a likelihood ratio of 6.04. Conclusions: Eosinophil count enables rapid routine screening of symptomatic chronic hemodialysis patients suspected of being COVID-19 within a range of low or high probability.

## 1. Introduction

Severe Acute Respiratory Syndrome related-Coronavirus 2 (SARS-CoV-2) infection, also called Coronavirus Disease-19 (COVID-19) is a viral infection caused by a ribonucleic acid (RNA) virus of the coronavirus family. Since it was first described in Wuhan, China, this disease has become a global pandemic, and by April 29th 2020 more than three million people had been infected and more than 200,000 had died.

Chronic dialysis patients are a vulnerable group at high risk of SARS-CoV-2 contamination with at least one comorbidity–such as hypertension, being elderly and diabetes–associated with COVID-19 mortality [1,2]. First, dialysis patients are overexposed to the risk of disease transmission for logistical reasons (regular presence at health care facilities, repeated trips by ambulance or taxi and physical proximity of patients during hemodialysis) and have difficulties with respect to social distancing. Second, it is essential to be able to quickly diagnose affected dialysis patients in order to prevent the spread of the disease within the ward and to protect the dialysis population in each center. The workflow of chronic dialysis patients can be quickly stretched in the context of COVID-19. The implementation of a clinical triage of patients upon their arrival in the dialysis center makes it possible to identify patients suspected of infection. Dialysis centers have therefore set up COVID-19 isolation zones to limit the risk of transmission to non-suspect patients while waiting for real-time reverse-transcriptase polymerase chain reaction (RT-PCR) results. Each suspension creates stress and puts a strain on the organization of the dialysis center waiting the final diagnosis [3,4].

Diagnosis is based on nasopharyngeal real-time RT-PCR, for which the feasibility and timeliness depend on the capacities of each center. In any case, however, this procedure does not permit classification of the patient as COVID-19 positive or negative in less than 4 h in addition with the risk of false negative [5]. Chest computed tomography (CT) scans can screen for low or high suspicion of COVID-19 but these are not available at all dialysis centers [6]. Some biological parameters, such as ferritin, lymphocyte and eosinophil count, have been studied for screening the patient for a low or high suspicion of COVID-19 but no studies have been conducted in dialysis patients to date [7]. Patients undergoing dialysis receive a weekly or monthly schedule of biological monitoring [8]. The results of a blood count and standard biochemical analyses are available in less than 2 h.

We hypothesize that anomalies in the biological report on the day of suspected COVID infection, compared to the monthly report for a patient on their arrival in the dialysis center, can be identified in order to quickly determine a low or high suspicion of COVID-19 in less than 2 h.

## 2. Methods

### 2.1. Setting

In the context of the global pandemic of SARS-CoV-2, we have set up clinical screening for SARS-CoV-2 infection when patients arrive at the dialysis center of the Hôpital de la Conception, Assistance Publique–Hôpitaux de Marseille (APHM), Marseille, France. We prospectively collected data from patients identified as suspects during this screening between March 12th and April 24th 2020. The suspected cases were all tested for the SARS-CoV-2 virus by nasopharyngeal real-time RT-PCR to determine whether they were COVID positive or negative. Positive RT-PCR were confirmed twice times. Presence of one of the following symptoms at arrival in the dialysis unit suggested SARS-CoV-2 infection: fever, cough, dyspnea, rhinorrhea, headache, asthenia, anosmia, ageusia, diarrhea, nausea and/or vomiting, myalgia, confusion. The data included in this study was anonymized, approved according to General Data Protection Regulation and registered at the Health Data Portal and Data Protection Commission of APHM under the references PADS-20-154 and 2020-58. The patients were provided with oral information about this study.

### 2.2. Participants

The inclusion criteria in the study were: nasopharyngeal real-time RT-PCR assay for SARS-CoV-2 infection and complete blood count (CBC) on the same day. The exclusions criteria were: age < 18 years, patients under corticosteroid treatment, chemotherapy within the last three months, recent acute stress (severe trauma, major surgery, epileptic seizure, myocardial infarction in the previous month) and active hematological disease. Patients who did not have a CBC on the previous routine monthly workup or whose initial nasopharyngeal real-time RT-PCR had not been analyzed at the APHM laboratory were excluded. The COVID-19 patients have been reported in another accepted publication [9].

### 2.3. Data Source/Measurement

#### 2.3.1. Epidemiological and Clinical Data

From electronic medical records we collected the following data: demographic, clinical, laboratory results, nucleic acid test results. Baseline patient characteristics were collected from electronic medical records: age, gender, body mass index (BMI), comorbidities (initial nephropathy, vascular access, history of immunosuppression or kidney transplantation, heart failure, coronaropathy, peripheral artery disease, arrhythmia, chronic respiratory disease, diabetes, cancers, hypertension and smoking) and their significant treatments such as angiotensin-converting enzyme inhibitors (ACEI), angiotensin receptor blockers (ARB), vitamin K antagonist, calcium channel blockers, beta blockers, aspirin, clopidogrel, statins, non-steroidal anti-inflammatory drugs (NSAIDs), iron supplementation and erythropoietin in dialysis.

#### 2.3.2. Laboratory Procedures

Methods for laboratory confirmation of SARS-CoV-2 infection: one virology laboratory was responsible for SARS-CoV-2 detection in respiratory specimens using real-time RT-PCR methods. Throat-swab specimens were obtained for SARS-CoV-2 RT-PCR in the dialysis unit. The system targeted the envelope protein (E)-encoding gene, as described previously [10]. RT-PCR was considered negative over a 34-cycle threshold (CT) value.

Routine blood examinations were CBC by an automated cell counter and serum biochemical tests (electrolyte, albuminemia, C-reactive protein [CRP]). We collected the routine monthly blood test monitoring (CBC, electrolyte, albuminemia, CRP) (results from March) for hemodialysis patients.

#### 2.3.3. Statistical Analysis

Continuous and categorical variables were presented as median (interquartile range [IQR]) and *n* (%), respectively. Sensitivity and specificity, as well as positive and negative predictive values, were calculated.

We used the Mann–Whitney U test, χ^2^ test, or Fisher’s exact test to compare differences between negative and positive COVID-19 where appropriate. All tests were two-tailed.

Unconditional logistic regression analysis was used to determine whether each variable was an independent factor in COVID-19 diagnosis. Covariates for the multivariate logistic regression analysis were selected based on a *p*-value < 0.05 in a univariate analysis. Variables were considered significant if *p* < 0.05, and the results are presented as odds ratio with 95% confidence intervals (CIs). Diagnostic accuracy for COVID-19 was assessed using the receiver operating characteristic area under curve (ROC AUC). Cut-off values showing the greatest accuracy were determined using sensitivity/specificity. All statistical analyses performed with the Prism 8 (GraphPad Software Inc., La Jolla, CA, USA).

## 3. Results

### 3.1. Patient Characteristics

60 patients were tested for SARS-CoV-2 RNA detection by nasopharyngeal swab at the Hôpital de la Conception, APHM. After excluding 23 patients according to the non-inclusion criteria, 37 were included in the final analysis (flowchart, Figure 1). Among the 37 patients included, 39 real-time RT-PCR tests were performed (2 patients were screened twice) with a peak at week 3 (Figure 2). 21 patients were negative for the SARS-CoV-2 RT-PCR and 16 were positive (Table 1). 22 RT-PCR were negative and 17 were positive. The median age was 72 years (IQR 54.5–79), with a median BMI of 23.3 kg/m^2^, and most of the patients were male (Table 1). Hypertension was the most represented comorbidity (86.5%), followed by atrial fibrillation (32.4%) and diabetes (30.6%) and only 4 patients had chronic respiratory disease (10.8%) (Table 1). Antihypertensive treatments, particularly ACEI and ARB, are detailed in Table 1. At baseline, corresponding to the March blood sample, patients had normal white blood cell count, and none of these clinical or biological data differed between the COVID-19 positive or negative patient groups (Table 1).

The day of suspected COVID-19, total leukocytes were significantly lower in the COVID-19 positive group (4.1 vs. 7.4 G/L *p* = 0.0072). The white blood cell count was characterized by lower neutrophils (2.7 vs. 5.1 G/L, *p* = 0.021) and eosinophils (0.01 vs. 0.15 G/L, *p* = 0.0003). The remaining biological variables were not significantly different (Table 2). Compared to their baseline biological status, neutrophils from the COVID-19 negative group increased significantly the day of the COVID-19 suspicion (3.6 vs. 5.1 G/L, *p* = 0.008). For the COVID-19 positive group, lymphocytes (1.2 vs. 0.8 G/L, *p* = 0.001) and eosinophils decreased significantly (0.18 vs. 0.01 G/L, *p* < 0.0001). In both groups, we observed an increase in CRP (6.2 vs. 13, *p* = 0.02; 2.3 vs. 34.2, *p* = 0.001) (Table 3).

### 3.2. Diagnostic Accuracy of Eosinopenia

Eosinopenia was observed in 14 out of 17 the SARS-CoV-2 RT-PCR positive group versus 3 out of 22 the SARS-CoV-2 RT-PCR negative group. ROC AUC for the detection of SARS-CoV-2 was 0.9 (0.81–1) (*p* < 0.0001). The highest diagnostic accuracy was observed for eosinophil count cut-off at 0.045 G/L. The eosinopenia diagnostic performance for SARS-CoV-2 infection showed a sensitivity of 82%, specificity of 86%, a positive predictive value of 82%, a negative predictive value of 0.86% and a likelihood ratio of 6.04 (Table 4 and Figure 3 and Figure 4).

## 4. Discussion

This is the first study to show that development of eosinopenia can differentiate low and high COVID-19 suspicion in chronic dialysis patients, with high diagnostic accuracy.

Eosinophils are found predominantly in tissues, with a smaller fraction found in circulation. The half-life of the eosinophil in the peripheral blood of normal individuals is approximately 18 h, with an average blood transit time of around 26 h, similar to that of neutrophils. Eosinophil is a cell that is principally present in extravascular sites in quantities several hundred times greater than those in peripheral blood. Circulating cells reflect only those that transit between the blood marrow and their final extravascular functional destination. During certain acute inflammatory or immune responses, a time lag between the migration of circulating eosinophils to the tissue where the immune response takes place and the induction of eosinophil synthesis and marrow emigration is observed [11]. This leads to the development of either eosinopenia or delayed blood eosinophilia, or both, and may explain the presence of eosinopenia in patients with COVID-19 disease. Our results indicate that early development of eosinopenia could reflect a powerful acute inflammatory and immune response triggered by SARS-CoV-2 infection. The role of eosinopenia in COVID-19 remains unclear and may be multifactorial. Whether the acquired eosinopenia associated with COVID-19 directly contributes to the disease course or is a marker of severe disease has not yet been determined. However, evaluation of the eosinophilic blood count represents a useful tool to manage early SARS-CoV-2 suspicion for the dialysis patients and in deciding to promptly isolate a patient from the other dialyzed patient in the center.

In this study, eosinophil count reliably discriminated between patients with and those without COVID-19 with an AUC of 0.9 by using a cutoff of 0.045 G/L within 24 h of the suspected diagnosis. The discrimination between low and high COVID-19 suspicion is a challenge and clinically relevant. We have not tested the role of eosinopenia in comparison with influenzae in our cohort because we did not observe co-infection in our center. This point has been studied by Andreozzi et al. in a letter to editor and raise the point that complete eosinopenia is a common finding in both COVID-19 and Seasonal Influenza infections. Eosinopenia is a potential biological indicator of either Influenza or SARS-COV-2 infections. However, complete eosinopenia should raise the suspicion of a COVID-19 infection outside of the flu season [12]. We believe that detection of eosinopenia is of interest to detect more quickly COVID-19 infection and promptly isolated the patient in the dialysis center. The ability to identify high COVID-19 suspicion with an inexpensive, widely available, point-of-care test has important practical implications, particularly in the efforts to screen hemodialysis patients during their thrice weekly management. Interestingly, classical markers of inflammation such as CRP are not discriminating in our population, as COVID negative patients were subject to an infectious process during screening. In COVID-19 dialysis patients, we found a similar tendency to lymphopenia and thrombocytopenia as in the general COVID-19 population [13]. In our study the onset of lymphopenia is non-discriminatory, probably because this population is characterized by acquired immune deficiencies secondary to the uremic stage. In contrast, we found no tendency to hypokalemia in our COVID-19 dialysis patients, which can be partly explained by end-stage renal disease.

Molecular biology and chest CT scans, if available, with subsequent results, take more than 12 h for most chronic hemodialysis centers. In contrast, CBC is a routine procedure in these centers. Results are obtained within one hour, allowing for the identification of low or high suspicion soon after the arrival of the dialysis patient. In our study, more than 50% of suspected patients included in the final analysis tested negative for COVID. The diagnostic approach was based on the result of the SARS-CoV-2 real-time RT-PCR, where reporting time did not permit ruling out or confirming the diagnosis of COVID-19 by the end of the dialysis session. The presence of eosinopenia could thus make it possible to classify patients as low or high suspicion and help the clinician to improve the diagnostic process.

Our study has a small sample size, which is its main limitation. We have deliberately excluded patients with factors that might have interfered with eosinophil interpretation and thus represent another limitation of our study. In addition, we benefited from pre-pandemic biological characteristics, allowing us to show the development of eosinopenia using the COVID-19 diagnostic. The diagnostic accuracy of our study needed to be externally validated in another data set of hemodialysis patient and represent a limitation of the study.

These encouraging results lead us to believe that it is possible to carry out a systematic screening of patients based on the CBC at each weekly check-up. It would be interesting to assess if eosinopenia enable to identify asymptomatic patients and reduce contagiousness in our vulnerable population.

## 5. Conclusions

To conclude, the detection of eosinopenia enables rapid triage of symptomatic chronic hemodialysis patients into low or high COVID-19 suspicion groups when they arrive at the dialysis session. The ability to identify high COVID suspicion with an inexpensive, widely available, point-of-care test, has important practical implications, particularly for early hemodialysis isolation to avoid spread of SARS-CoV-2 throughout a center. This low cost triage tool is of particular interest in the coming months, especially for low-income countries with limited access to RT-PCR and chest CT scans [14,15].

## Figures and Tables

**Figure 1 jcm-10-00004-f001:**
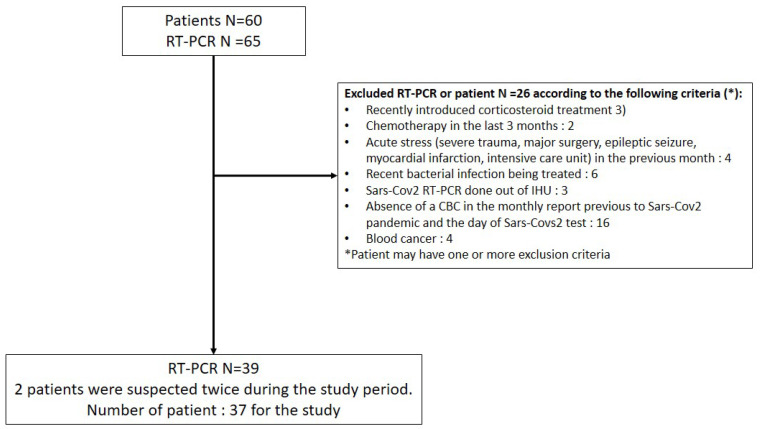
Flowchart. IHU, Institut Hospitalo-universitaire-Méditerranée Infection; RT-PCR, reverse-transcriptase polymerase chain reaction.

**Figure 2 jcm-10-00004-f002:**
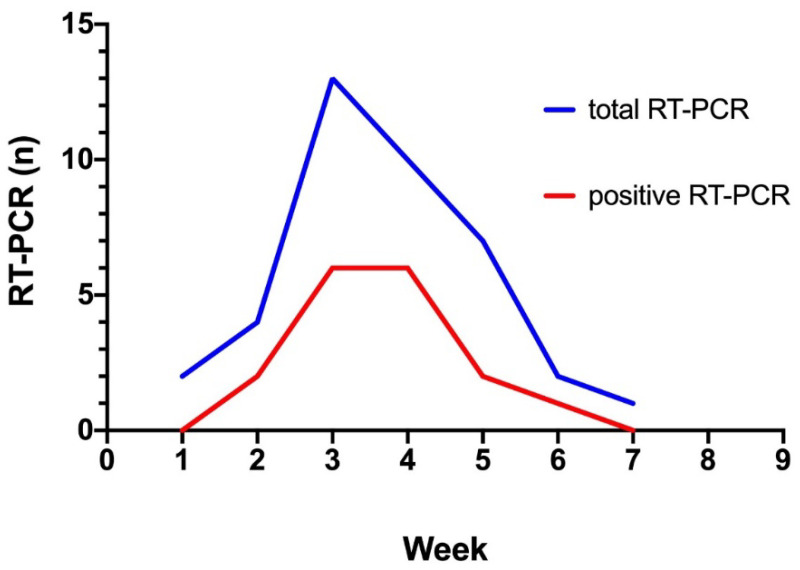
Evolution of the number of reverse-transcriptase polymerase chain reaction (RT-PCRs) testing for SARS-CoV-2 performed between March 12 and April 24 2020. The blue line represents the evolution of the number of RT-PCR performed between week 1 and 8. The red line represents the evolution of the number of positive RT-PCR sent during week 1 and 8.

**Figure 3 jcm-10-00004-f003:**
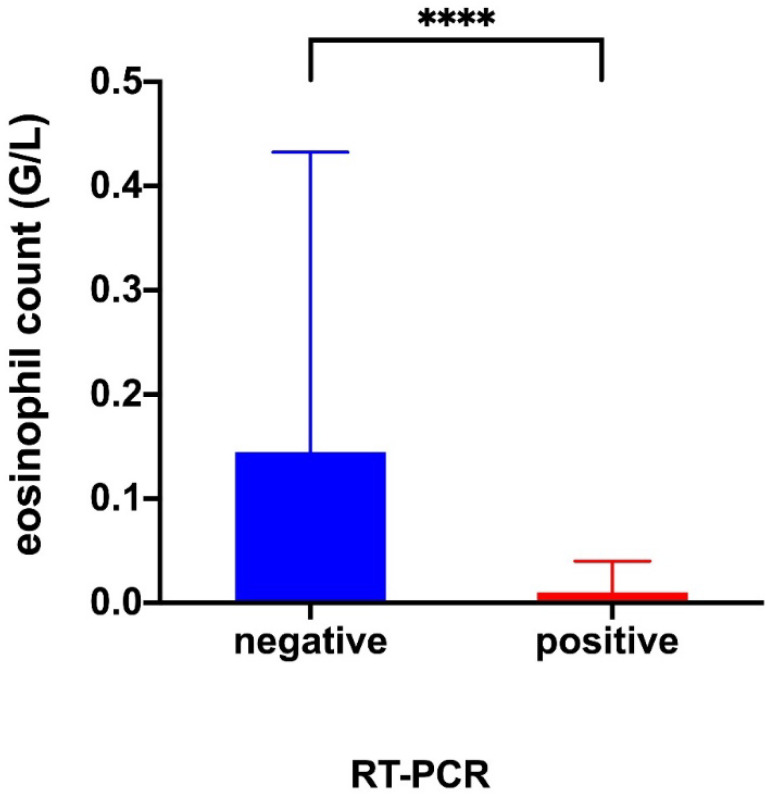
Comparison of the eosinophil level when performing a cell blood count on the day of a COVID-19 suspicion. Values represented are median and IQR. Mann–Whitney test. RT-PCR, reverse-transcriptase polymerase chain reaction; ****, represent *p* value < 0.0001.

**Figure 4 jcm-10-00004-f004:**
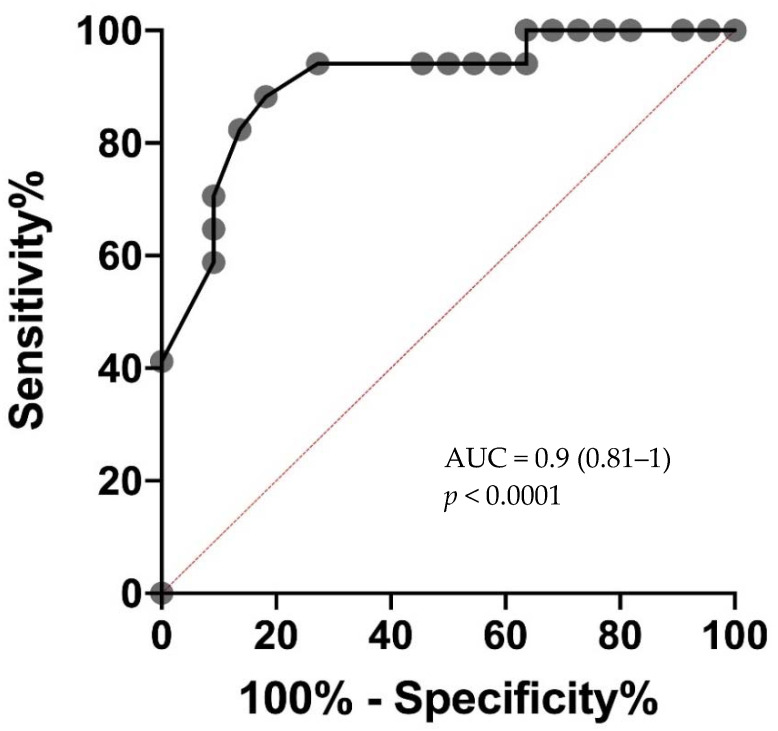
Receiver operating characteristic (ROC) curve of eosinophil count showing specificity and sensitivity for COVID-19 diagnosis. AUC, area under the ROC curve.

**Table 1 jcm-10-00004-t001:** Patients baseline characteristics.

Variable ^a^	Total *n* = 37	Negative COVID *n* = 21	Positive COVID *n* = 16	*p* Value
Male	24 (64.9)	12 (57.2)	12 (75)	0.31
Age (years)	72 (54.5–79)	72 (48–83.5)	71 (55–80.9)	0.79
Weight (kg)	63.5 (57.5–70.6)	64.5 (58–69.3)	63.5 (55.6–81.8)	0.32
BMI (kg/m^2^)	23.3 (20.6–24.6)	23.7 (20.8–24.5)	23.3 (19.8–25.7)	0.92
**Nephropathy:**				-
Glomerular	11 (29.7)	5 (23.8)	6 (37.5)	
Vascular	8 (21.6)	5 (23.8)	3 (18.6)	
Tubular	8 (21.6)	7 (33.3)	1 (6.3)	
Genetic	2 (5.5)	1 (4.8)	1 (6.3)	
Not determined	8 (21.6)	3 (14.3)	5 (31.3)	
**Vascular access:**				
Fistula	26 (70.3)	15 (71.4)	11 (68.8)	1
Central catheter	11 (29.7)	6 (28.6)	5 (31.2)	
Immunosuppression	11 (29.6)	7 (33.3)	4 (28.6)	1
History of graft kidney	7 (18.9)	4 (19.1)	3 (20)	1
**Comorbidities**				
Hypertension	32 (86.5)	19 (90.5)	13 (81.3)	0.63
Congestive heart failure	6 (12.5)	4 (19.5)	2 (12.5)	0.68
Coronary heart disease	7 (18.9)	2 (9.5)	5 (31.2)	0.2
Peripheral vascular disease	7 (18.9)	3 (14.3)	4 (25)	0.44
Cardiac arrhythmia	12 (32.4)	8 (40)	4 (25)	0.48
Chronic respiratory disease	4 (10.8)	3 (14.3)	1 (6.3)	0.62
Diabetes	11 (30.6)	7 (35)	4 (25)	0.72
Cancer	4 (10.8)	1 (4.8)	3 (18.8)	0.3
Smoker	5 (13.9)	4 (19.5)	1 (6.3)	0.35
**Medication:**				
ACE inhibitors	6 (16.2)	2 (9.5)	4 (25)	0.17
ARBs	5 (13.5)	3 (14.3)	2 (12.5)	1
Beta blocker	12 (32.4)	6 (28.6)	6 (37.5)	0.73
Calcium channel blockers	10 (27.0)	6 (28.6)	4 (25)	1
Diuretic	1 (2.7)	0	1 (6.3)	0.43
Aspirin	12 (32.4)	7 (33.3)	5 (31.3)	1
Clopidogrel	3 (8.1)	2 (9.5)	1 (6.3)	1
VK	9 (24.3)	6 (28.6)	3 (18.8)	0.7
Statin drug	5 (13.5)	1 (4.8)	4 (25)	0.14
Steroids	0	0	0	-
ASEs	27 (73)	17 (61.9)	10 (38.1)	0.38
**March biological values**				
Leukocyte (G/L)	5.9 (4.6–6.5)	5.8 (4.6–7.9)	5.9 (4.7–6.2)	0.54
Neutrophil (G/L)	3.6 (2.8–4.6)	3.6 (3.1–5.3)	3.6 (2.4–4.2)	0.44
Lymphocyte (G/L)	1 (0.75–1.45)	1 (0.6–1.2)	1.2 (0.9–1.6)	0.16
Monocyte (G/L)	0.6 (0.4–0.8)	0.6 (0.45–0.9)	0.5 (0.4–0.7)	0.29
Eosinophil (G/L)	0.17 (0.01–0.33)	0.17 (0.1–0.3)	0.18 (0.11–0.35)	0.44
Platelet (G/L)	189 (143–249)	198 (149–258)	179 (138–235)	0.71
Hemoglobin (g/dL)	11.2 (10.4–11.6)	11.2 (10–11.6)	11.2 (10.6–11.8)	0.78
CRP (mg/L)	4.7 (1.35–8.45)	6.2 (2.8–9.9)	2.3 (0.7–38.8)	0.95
Albumin (g/L)	39.3 (37.2–42)	39 (37.2–42)	39.8 (37.2–42.3)	0.47
Potassium (mmol/L)	4.5 (4.1–5.3)	4.3 (4–5)	5.1 (4.1–5.7)	0.2

^a^ For quantitative variables, values are expressed as median (interquartile range). For qualitative variables, values are expressed as *n* (%). ACE, angiotensin-converting enzyme; ARBs, angiotensin-receptor blockers; ASEs, Erythropoiesis-stimulating agents; BMI, body mass index; CRP, C-reactive protein; VK, vitamin K antagonist; -, No statistic test were performed.

**Table 2 jcm-10-00004-t002:** Variables the day of COVID-19 suspicion.

Variables ^a^	Negative COVID	Positive COVID	*p* Value
Leukocyte (G/L)	7.4 (4.9–10.4)	4.1 (3.3–7.1)	0.0072
Neutrophil (G/L)	5.1 (3.3–8.1)	2.7 (2.2–5.6)	0.021
Lymphocyte (G/L)	0.85 (0.57–1.22)	0.8 (0.55–1.05)	0.29
Monocyte (G/L)	0.80 (0.47–0.92)	0.5 (0.3–0.8)	0.099
Eosinophil (G/L)	0.15 (0.06–0.43)	0.01 (0–0.04)	0.0003
Platelet (G/L)	201 (152–255)	162 (118–185)	0.077
Hemoglobin (g/dL)	11.1 (10.5–11.6)	11.2 (10.4–12.4)	0.81
CRP (mg/L)	13 (3.3–65.5)	34.2 (15.9–72.8)	0.57
Albumin (g/L)	40.5 (36.1–42.4)	37.6 (34.2–41.5)	0.24
Potassium (mmol/L)	4.6 (4.15–5.55)	4.8 (4.1–5.1)	0.32

^a^ For quantitative variables, values are expressed as median (interquartile range). CRP, C-reactive protein.

**Table 3 jcm-10-00004-t003:** Biological comparison between March monitoring and the suspicion day.

Variable ^a^	Negative RT-PCR		*p* Value	Positive RT-PCR		*p* Value
	Monthly Assessment	Suspicion Day		Monthly Assessment	Suspicion Day	
Leukocyte (G/L)	5.8 (4.6–7.9)	7.4 (4.9–10.4)	0.09	5.9 (4.7–6.2)	4.1 (3.3–7.1)	0.16
Neutrophil (G/L)	3.6 (3.1–5.3)	5.1 (3.3–8.1)	0.008	3.6 (2.4–4.2)	2.7 (2.2–5.6)	0.84
Lymphocyte (G/L)	1 (0.6–1.2)	0.85 (0.57–1.22)	0.30	1.2 (0.9–1.6)	0.8 (0.55–1.05)	0.001
Monocyte (G/L)	0.6 (0.45–0.9)	0.8 (0.47–0.92)	0.48	0.5 (0.4–0.7)	0.5 (0.3–0.8)	0.82
Eosinophil (G/L)	0.17 (0.1–0.3)	0.15 (0.06–0.43)	0.23	0.18 (0.11–0.35)	0.01 (0–0.04)	<0.0001
Platelet (G/L)	198 (149–258)	201 (152–255)	0.64	179 (138–235)	162 (118–185)	0.004
Hemoglobin (g/dL)	11.2 (10–11.6)	11.1 (10.5–11.6)	0.24	11.2 (10.6–11.8)	11.2 (10.4–12.4)	0.71
CRP (mg/L)	6.2 (2.8–9.9)	13 (3.3–65.5)	0.02	2.3 (0.7–38.8)	34.2 (15.9–72.8)	0.001
Albumin (g/L)	39 (37.2–4)	40.5 (36.1–42.4)	0.38	39.8 (37.2–42.3)	37.6 (34.2–41.5)	0.07
Potassium (mmol/L)	4.3 (4–5)	4.6 (4.15–5.6)	0.16	5.1 (4.1–5.7)	4.8 (4.1–5.1)	0.27

^a^ For quantitative variables, values are expressed as median (interquartile range). CRP, C-reactive protein; RT-PCR, reverse-transcriptase polymerase chain reaction.

**Table 4 jcm-10-00004-t004:** Diagnostic performance for eosinopenia and RT-PCR the day of suspicion.

Effect Size	Value	95% CI
Sensitivity	0.82	0.59 to 0.94
Specificity	0.86	0.67 to 0.95
Positive Predictive Value	0.82	0.59 to 0.94
Negative Predictive Value	0.86	0.67 to 0.95
Likelihood Ratio	6.04	

CI, confidence interval; RT-PCR, reverse-transcriptase polymerase chain reaction.

## Data Availability

The data presented in this study are available on request from the corresponding author T.R.

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
