# Peer review of "Triage of Patients Suspected of COVID-19 in Chronic Hemodialysis: Eosinophil Count Differentiates Low and High Suspicion of COVID-19"

_jcm, 2020, doi:10.3390/jcm10010004_

Round 1

Reviewer 1 Report

The authors described the importance of eosinophilia to distinguish dialysis patients suspected of Covid-19. The finding seems to support other reports that also emphasizing eosinophil count as a parameter to suspect Covid-19 patients. Although the sample number is small, but the authors already acknowledge this in the paper.

Author Response

We thank you the reveiwer for the valuable comment.

Reviewer 2 Report

This manuscript tries to find a reliable and quickly accessible diagnostic indicator to improve the suspicion of COVID-109 among hemodialysis patients attending the dialysis center of the Hôpital de la Conception, Hôpitaux de Marseille.

The authors found that an eosinophil count below 0.045 G/L identified SARS-CoV-2 infection with AUC of 0.9, sensitivity of 82% and specificity of 86%. Although these findings can be relevant for nephrologists that have to reduce the transmission of the virus among patients, the small sample size does not allow to generalize these data. The idea to perform CBC along with PCR for SARS-Cov-2 is interesting but now we cannot rely on diagnostic tools with high false negative rates for testing.

Author Response

We thank the reviewer for this interesting remark. However, we believe that this result is particularly interesting for chronic dialysis centers treating patients 3 times a week. A systematic check-up including a blood count is performed on a weekly basis. The appearance of eosinopenia would make it possible to be more vigilant and to perform a nasopharyngeal PCR more rapidly, particularly in asymptomatic patients.

Reviewer 3 Report

This is nice and smart work, showing the ability to triage hemodialysis patients into low or high COVID-19 suspicion groups at the dialysis session, using complete blood count. Eosinophil count cut-off at 0.045 was helpful in recognizing COVID infection with 82% sensitivity and 86% specificity in these patients.

Although the problem is impressive, the article contains some flaws.

Major comments:

  1. RT-PCR test was performed only once; thus, false-negative results were not verified. Moreover, tests were performed in the COVID pandemic time, and SARS-CoV-2 infections occur more frequently. In addition, the role of eosinopenia was not tested in comparison to other often co/infections (for example with influenza). Thus, eosinopenia is probably nonspecific for exclusive SARS-CoV-2 disease, which decreases the significance of this study. The over-mentioned problems are other limitations of the presented work.
  2. In lines 132-134, the Authors wrote: " 21 patients were negative for the SARS-CoV-2 RT-PCR and 16 were positive (Table 1). 22 RT-PCR were negative and 17 were positive." Which sentence is the correct one?
  3. Figure 2 (lines 158-162) does not correspond to its description in lines 151-152: "The white blood cell count was characterized by lower neutrophils (2.7 vs 5.1 G/L, p=0.021) and eosinophils (0.01 vs 0.15 G/L, p=0.0003) (Figure 2)" - This is confusing and has to be corrected.

Minor comments:

  1. The role of eosinopenia could be more widely described.
  2. References contain French-language phrases.

Author Response

Response to Reviewer’s COMMENTS :

MAJOR COMMENTS

  1. RT-PCR test was performed only once; thus, false-negative results were not verified. Moreover, tests were performed in the COVID pandemic time, and SARS-CoV-2 infections occur more frequently. In addition, the role of eosinopenia was not tested in comparison to other often co/infections (for example with influenza). Thus, eosinopenia is probably nonspecific for exclusive SARS-CoV-2 disease, which decreases the significance of this study. The over-mentioned problems are other limitations of the presented work

Thomas ROBERT : We thank the rewievers for raising this point which is not clear enough in the manuscript.

All the patient have more than 2 PCR confirmations and we added this relevant point in the method section.

We agree that we have not tested the role of eosinopenia in comparison with influenzae in our cohort because we did not observed co-infection in our center. This point have been studied by Andreozzi et al in a letter to editor and raise the point that complete eosinopenia is a common finding in both COVID-19 and Seasonal Influenza infections. Eosinopenia is a potential biological indicator of either Influenza or SARS-COV-2 infections. But complete eosinopenia should raise the suspicion of a COVID-19 infection outside of the flu season.We believe that our work is of interest for the nephrologist and the dialysis patients to detect more quickly COVID-19 infection and promptyl isolated the patient in the dialysis center.

  1. In lines 132-134, the Authors wrote: " 21 patients were negative for the SARS-CoV-2 RT-PCR and 16 were positive (Table 1). 22 RT-PCR were negative and 17 were positive." Which sentence is the correct one?

Thomas ROBERT : We thank you the reviewer for this comment. The 2 sentences are correct because we mentioned that 2 patients were tested twice on line 132 explaining our result with 37 patients and 39 PCR.

  1. Figure 2 (lines 158-162) does not correspond to its description in lines 151-152: "The white blood cell count was characterized by lower neutrophils (2.7 vs 5.1 G/L, p=0.021) and eosinophils (0.01 vs 0.15 G/L, p=0.0003) (Figure 2)" - This is confusing and has to be corrected.

Thomas ROBERT : We thank you the reviewer for this valuable correction. We have suppress in the text Figure 2.

MINOR COMMENTS

  1. The role of eosinopenia could be more widely described.

Thomas ROBERT : We thank you the reviewer for this suggestion.We added in the discussion section the following sentence : The role of eosinopenia in COVID-19 remains unclear and may be multifactorial. Whether the acquired eosinopenia associated with COVID-19 directly contributes to the disease course or is a marker of severe disease has not yet been determined. However, evaluation of the eosinophilic blood count represents a useful tool to manage early SARS-CoV-2 suspicion for the dialysis patients and in deciding to promptly isolate a patient from the other dialysed patient in the center.

  1. References contain French-language phrases.

Thomas ROBERT : We thank you the reviewer for this point. We have corrected them in english.

Round 2

Reviewer 2 Report

The scientific message is novel but the utility of eosinopenia is limited by a poor specificity.

Author Response

  1. The scientific message is novel but the utility of eosinopenia is limited by a poor specificity.

Thomas ROBERT : We thanks the reviewer for this comment. We would like to moderate its conclusion because the specifity is higher than 82% with a highly informative AUC. Furthermore, the likelihood ratio is higher than 5 which is very informative to consider high suspicion of SARS-CoV-2 infection and making the eosinopenia parameter a usefullness tool with a real impact in the clinical practice in hemodialysis center. Noteworthy, we agreed that RT-PCR remain the gold standard for the SARS-CoV-2 infection dianostic.